# Encouraging Critical Thinking for Multi-Agent Debate

## Abstract

Large language models (LLMs) have demonstrated remarkable performance across a wide range of tasks in recent years. While prior work has explored leveraging LLMs to generate synthetic data for self-improvement, repeated iterations often suffer from diminishing returns due to the reliance on homogeneous reasoning patterns and limited exploration of alternative perspectives. In this paper, we introduce a novel framework that enriches the reasoning process by encouraging critical thinking among multiple agents. Rather than deploying an ensemble of models with identical prompts, we propose a *strategy generator* that produces customized instructions tailored to each individual LLM. Acting as a critical thinking agent, the generator is iteratively fine-tuned using carefully selected strategies that are both diverse and effective. This approach fosters specialization within each model while promoting diversity across reasoning paths, enabling the system to maintain varied solution trajectories and achieve sustained performance gains through iterative refinement. We demonstrate the effectiveness of our method across a variety of agentic frameworks and complex reasoning tasks.

## 1 Introduction

In recent years, Large Language Models (LLMs) have experienced unprecedented advancements in domains such as language generation, comprehension, question answering, and translation (Touvron et al., 2023; Chowdhery et al., 2023; Achiam et al., 2023; OpenAI, 2024). These advancements are largely due to research efforts focused on the reasoning process (Wei et al., 2022; Wang et al., 2022; Yao et al., 2023; Besta et al., 2024; Gao et al., 2024). While logical chains have been significantly enhanced, LLMs continue to produce incorrect statements that conflict with their original claims. To address this issue, research on self-reflection has been proposed to improve consistency by evaluating and refining initial responses (Madaan et al., 2023; Kim et al., 2023; Shinn et al., 2023). However, the improvements become marginal with deeper self-reflection and multiple rounds of fine-tuning (Subramaniam et al., 2025). Additionally, constrained by homogeneous reasoning, these methods struggle to effectively correct mistakes. Without an external strategy as guidance, self-reflection ultimately leads to diminished performance (Huang et al., 2023).

One effective approach to addressing homogeneous reasoning is to encourage fine-tuning across multiple models using subsets of the dataset, promoting both specialization and diversification in responses. To ensure the quality of data for fine-tuning, a multi-agent debate (MAD) mechanism is employed to generate robust pseudo-labels (Du et al., 2023). An alternative way to conceptualize this issue is through the lens of a "mental set"—a cognitive bias that hinders the ability to explore diverse approaches, particularly when faced with novel or more complex tasks (Öllinger et al., 2008). Building on this, researchers have proposed the Diverse Multi-Agent Debate framework, which guides LLMs using predefined and varied reasoning strategies (Liu et al., 2025). As a result, the use of unique prompting strategies fosters divergent thinking and improves problem-solving capabilities.

Despite the success of prior approaches, we argue that predefined strategies are not always accessible and may fail to cover optimal solution paths. Moreover, customizing fine-tuning for each individual LLM agent incurs substantial computational overhead. To address these challenges, we propose to foster critical thinking through a novel *strategy generator* within a multi-agent debate framework—**Critical Thinking with Multi-Agent Debate (CMAD)**. The core innovation of the proposed method lies in treating problem-solving strategies as entirely undefined and fully optimizable. This

unconstrained formulation enables broad exploration of the solution space, allowing the model to discover novel and potentially more effective strategies. However, the absence of structure introduces a high risk of failure due to the stochastic nature of exploration. To mitigate this, we optimize the strategy space to converge toward a sweet point between exploration, which promotes diversity and creativity, and exploitation, which ensures solution quality and reliability.

More specifically, given a question, the strategy generator is encouraged to produce diverse yet undefined strategies for solving the problem. Each distinct strategy is assigned to an independent agent, which then generates a solution conditioned on its assigned strategy. These agents subsequently engage in critique and evaluation of each other's solutions following established debate frameworks (Du et al., 2023; Liu et al., 2015). Notably, such strategy-conditioned generation promotes more structured outputs and effectively mitigates emerging cases where LLMs get lost in multi-turn conversations (Laban et al., 2025). While the initial strategies produced by the generator may be sub-optimal, we aim to iteratively improve its capability through a feedback loop grounded in two key perspectives: the `correctness` of final answers and the `diversity` of reasoning pathways. To assess correctness, we construct pseudo-labels by aggregating consensus outcomes from the multi-agent debate and multi-strategy evaluations. To measure diversity, we quantify the uniformity of the generated strategies. These two metrics are then used to guide sample selection, which in turn is used to fine-tune the strategy generator.

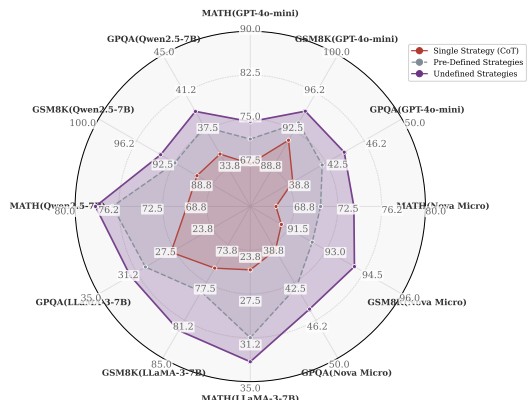

Figure 1: Undefined and optimizable strategies enhance problem-solving performance compared to fixed single-strategy (Wei et al., 2022) and predefined-strategy baselines. Predefined strategies are instantiated from classic reasoning approaches including Chain-of-Thought Prompting, Step-Back Prompting, and Program-of-Thoughts Prompting(Liu et al., 2025)

This feedback mechanism fosters the emergence of novel, specialized reasoning strategies and drives continuous improvement in LLM performance.

We quantitatively validate the effectiveness of the approach across a diverse set of reasoning tasks and LLMs, demonstrating consistent performance gains. The framwork is model-agnostic and integrates seamlessly with both open-source LLMs—such as `Qwen2.5` and `LLaMA-3`—and proprietary systems like `GPT-4o-mini` and `Nova Micro`, yielding marked improvements in solution quality. As shown in Figure 1, leveraging undefined and dynamically optimizable strategies within the critical thinking framework leads to significantly enhanced problem-solving capabilities, outperforming both fixed single-strategy baselines and predefined strategy paradigms. Furthermore, performance improves steadily with additional rounds of fine-tuning, demonstrating the scalability and robustness of the proposed framework.

Our main contributions are summarized as follows: (1)We propose a novel framework that encourages critical thinking in LLM agents by enabling them to generate diverse and undefined reasoning strategies, guided by a strategy generator. (2)We introduce a comprehensive feedback loop that evaluates both the correctness and diversity of agent responses, providing reliable, dynamic, and specialized guidance to LLMs with minimal computational overhead. (3) We empirically demonstrate that our fine-tuning paradigm for the strategy generator effectively encourages critical thinking and generalizes robustly across a wide range of datasets and popular LLMs.

## 2 RELATED WORK

**Multi-Agent LLM Reasoning:** Multi-agent LLM reasoning enhances performance by enabling interaction and collaboration among multiple language model agents (Liang et al., 2023; Wang et al., 2023a; Khan et al., 2024; Chan et al., 2023). To facilitate more effective communication and coordination, prior work has explored role assignment strategies to specialize agent behaviors (Liang

et al., 2023; Wang et al., 2023b; Chan et al., 2023). Another line of research encourages agents to challenge each other through iterative rounds of debate, promoting deeper reasoning and error correction (Du et al., 2023; Khan et al., 2024). While most debate-based frameworks treat agents as equally important participants, recent efforts have investigated expert-guided collaboration via meta-programming, inter-agent consistency, latent embeddings, and pre-defined reasoning paths (Hong et al., 2023; Xiong et al., 2023; Pham et al., 2023).

**Fine-tuning for Self-improvement:** Fine-tuning has been widely adopted to improve the performance of LLMs (Welleck et al., 2022; Hsieh et al., 2023; Huang et al., 2022; Subramaniam et al., 2025; Zhang et al., 2024b). Considerable fine-tuning methods aim to optimize models using prior data to encourage strategy generation through self-iterated learning (Pang et al., 2024; Anthony et al., 2017; Polu et al., 2022; Parthasarathy et al., 2024). In addition, reinforcement learning has emerged as a popular self-training technique, often demonstrating better generalization (Chen et al., 2024b;b;a). Notably, most of these approaches rely on ground-truth data. In contrast, we diverge from these paths by employing unsupervised multi-agent interaction to achieve more consistent performance gains, following recent research (Subramaniam et al., 2025). More importantly, we emphasize the role of diversity in the sample selection process—an aspect that is frequently overlooked in prior work.

**Critical Thinking for LLMs:** Critical thinking is a powerful capability for promoting error correction and uncovering inconsistencies. It has been employed to detect noncompliance in statements (Kamath et al., 2020; Brahman et al., 2024), identify knowledge conflicts and misinformation, and reveal inconsistencies in problem framing (Xie et al., 2023; Zhou et al., 2023; Xu et al., 2023; Chen & Shu, 2024). More recently, critical thinking has been adopted to enhance the reasoning capabilities of LLMs. This includes incorporating diverse reasoning paths, leveraging self-correction mechanisms (Tyen et al., 2023; Huang et al., 2023), quantifying reasoning quality through step-by-step scoring (Golovneva et al., 2022), and evaluating performance on specialized reasoning benchmarks (Zeng et al., 2024). Our work deviates from prior efforts by applying critical thinking to explore and enrich solution batches without pre-defined constraints on the reasoning process.

## 3 METHOD

While sticking to a single approach or a narrow set of strategies may lead to sub-optimal or dead ends, switching to unexplored methods can sometimes resolve challenging tasks more effectively. In this section, we present the framework for enabling diversified and critical thinking *without relying on pre-defined strategies*. We begin with an overview of the proposed approach in Section 3.1, illustrating how strategy generation guides the multi-agent system to complete a task. Section 3.2 then details the pipeline for extracting high-quality strategies from multi-agent interactions for fine-tuning.

### 3.1 OVERALL FRAMEWORK

We present the overall workflow in Figure 2. Given a task $x$ sampled from questions set $\mathcal{P}_q$ expressed in natural language, a strategy generator $G(x)$ takes $x$ as input and proposes a set of high-level strategies that could potentially solve the task. We define this strategy generation process as:

$$S_1, S_2, \ldots, S_M = G(x), \qquad x \sim \mathcal{P}_q \tag{1}$$

Based on this, we denote the generated strategy set as $\mathbb{S} = \{\mathcal{S}_i \mid i = 1, 2, \ldots, M\}$. Correspondingly, we initialize $M$ LLM agents, denoted as $\mathbb{A} = \{\mathcal{A}_i \mid i = 1, 2, \ldots, M\}$. In the first round of debate, each agent $\mathcal{A}_i$ is assigned a strategy $\mathcal{S}_i$ and tasked with generating a reasoning trajectory and a final answer, denoted as $y_{1,i}$, where the first subscript corresponds to the agent index and second indicates the debate round. Formally, the generation process in the first round is defined as:

$$y_{i,1} = \mathcal{A}_i(x; \mathcal{S}_i), \qquad i = 1, 2, \ldots, M. \tag{2}$$

In subsequent rounds, the responses and reasoning traces from the first round,i.e., $y_{1,1}, y_{2,1}, \ldots, y_{M,1}$, are aggregated into a shared historical context $h_1$, following the paradigm established in (Du et al., 2023). This shared history is then made available to all agents $\mathcal{A}i$. Conditioned on this history, the agents generate their second-round responses. This process is repeated iteratively in the following rounds. We define the general formulation in the $n^{th}$ as:

$$y_{i,n} = \mathcal{A}_i(x; h_{n-1}), \qquad i = 1, 2, \ldots, M, \quad n = 2, 3, \ldots, N. \tag{3}$$

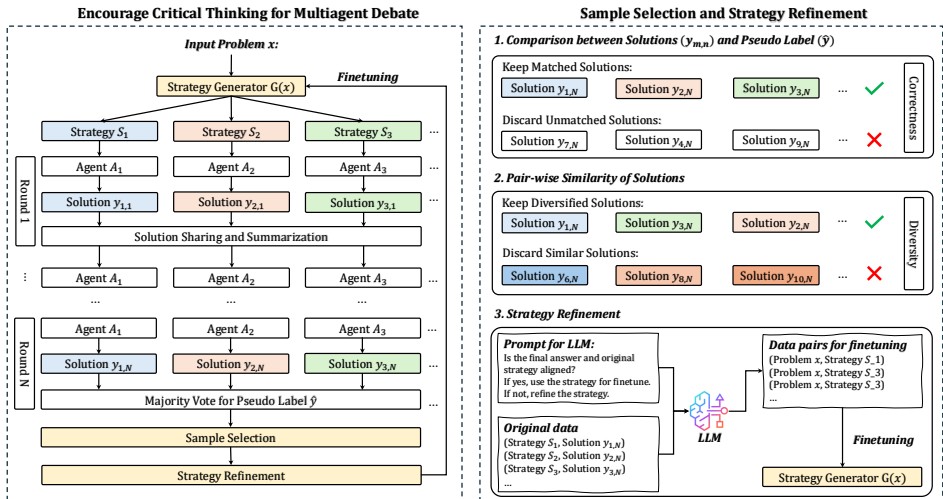

Figure 2: Overall Framework of Critical Thinking for Multi-Agent Debate. We first use strategy generation to guide each agent toward proposing diverse solutions. A majority vote is applied over the final-round answers to construct pseudo labels for fine-tuning (left). These pseudo labels help identify correct answers and effective strategies. Next, we evaluate the diversity of the generated solutions to further refine the pre-training data, improving the fine-tuning of the strategy generator and encouraging more diversified and reliable outputs (right). This figure illustrates a single fine-tuning iteration; applying multiple iterations can lead to further performance improvements.

## 3.2 FINETUNING STRATEGY GENERATOR WITH SELECTED SAMPLES

To empower the strategy generator with critical thinking, we first carefully choose samples that actively guide the generator to solve the given problem $x$ in correct and diverse ways. Correspondingly, we need metrics to quantify the correctness and diversity without knowledge of ground truth.

For correctness evaluation, we select the majority vote from the final round of debate across $M$ agents and $N$ rounds as the pseudo label $\hat{y}$. We then construct the dataset $\mathcal{D}_c$, consisting of samples whose solutions are aligned with $\hat{y}$, formally defined as:

$$\mathcal{D}_c \leftarrow \{y_{m,N} \mid y_{m,N} = \hat{y}, m \in \{1, 2, \ldots, M\}\} \tag{4}$$

While these pseudo labels are reliable with solution sharing among agents, the debate process inevitably leads to convergence toward similar reasoning trajectories. As a result, the final-round responses $y_{i,N}$ tend to follow closely aligned solution paths, producing increasingly similar outputs. This convergence effect ultimately leads to diminishing returns in problem-solving performance as the number of agents and debate rounds increases.

To further encourage diverse critical thinking, we introduce a diversity metric that evaluates pairwise similarity among generated solutions. The metric is designed to satisfy three key properties: (1) asymptotic correctness—converging to a uniform distribution to ensure maximum diversity, (2) empirically effective with finite samples, and (3) the ability to capture non-linear semantic relationships. To this end, we adopt the Gaussian potential kernel (RBF kernel) (Cohn & Kumar, 2007; Borodachov et al., 2019), defined as:

$$G_t(u, v) = e^{-t\|u-v\|^2} = e^{-t(\|u\|^2 + \|v\|^2 - 2u^\top v)}, \quad t > 0, \tag{5}$$

where $u, v \in \mathbb{R}^d$ are points in a $d$-dimensional Euclidean space, and $t$ is a temperature parameter. Then, we define the diversity metric $D$ as the logarithm of the expected pairwise Gaussian potential:

$$D(f; t) = \log \mathbb{E}_{x,y \sim \mathcal{D}_c} [G_t(f(x), f(y))] = \log \mathbb{E}_{x,y \sim \mathcal{D}_c} \left[ e^{-t\|f(x)-f(y)\|^2} \right], \quad t > 0, \tag{6}$$

where $x$ and $y$ represent independent and identically distributed samples from $\mathcal{D}_c$, and $f$ represents an embedding model.

Given the diversity metric, we proceed to select a subset of correct solutions. We start by including the first solution from our correct set $\mathcal{D}_c$, then systematically add solutions that are sufficiently different from those already selected. Specifically, for each candidate solution, we compute its pairwise similarity with all previously selected solutions and only include it if the maximum similarity is below a predefined threshold $\tau$. This ensures each new addition contributes novel reasoning approaches. Formally, we construct our filtered dataset as:

$$\mathcal{D}_{div} \leftarrow \{y_1\} \cup \{y_i \in \mathcal{D}_c \setminus \{y_1\} \mid \max_{y_j \in \mathcal{D}_{div}} |G_t(f(y_i), f(y_j))| > \tau\} \tag{7}$$

where $\tau$ is the similarity threshold that controls the diversity level of selected samples. A larger $\tau$ enforces greater diversity but potentially reduces the number of available training examples. This selection approach ensures that our strategy generator learns from a set of solutions that are not only correct but also represent diverse approaches to the same problem, thereby enhancing its critical thinking capabilities across various reasoning paths.

Finally, we introduce an optional strategy refinement stage for alignment between the concrete solution from the last round of debate $y_{i,N}$ and initial strategy $\mathcal{S}_i$. Specifically, we introduce another strategy alignment agent $\mathcal{A}_{\text{align}}$, with prompt $P_{\text{ref}}$ to identify and calibrate the logic of strategy $\mathcal{S}_i$ with the pseudo ground truth solutions $y_{i,N}$. The details of the prompt can be found in the D. Formally, we define the process as:

$$\hat{S}_i = \mathcal{A}_{\text{align}}(S_i, P_{\text{ref}}; \hat{y}_i), \qquad i = 1, 2, \ldots, M. \tag{8}$$

Thereby, we are able to construct a high-quality dataset. For each question $x$, we apply $D_f(x, \hat{S}_i)$ for the fine-tuning task of the strategy generator. We illustrate the comprehensive summary of the procedures in Algorithm 1.

---

**Algorithm 1** Critical Thinking Algorithm

---

**Require:** A set of input questions $\mathcal{P}_q = \{x_t\}$; The strategy generator $G(x)$; $M$ model instances $\{\mathcal{A}_i | i = 1, 2, ..., M\}$; The number of debate rounds $N$; The number of finetuning iterations $L$; The diversity threshold $\tau$;
1: Initialize dataset $\mathcal{D}_f$ for finetuning
2: **for** $l = 1 \rightarrow L$ **do**
3:      **for** $x$ in $\mathcal{P}_q$ **do**
4:          **for** Round $j = 0, ..., N$ **do**
5:              $S_1, \ldots, S_M \leftarrow G(x)$          ▷ Generate Strategies (Eq. 1)
6:              **if** $j = 0$ **then**
7:                  $y_{1,1}, \ldots, y_{M,1} \leftarrow \mathcal{A}_i(x; \mathcal{S}_1), \ldots, \mathcal{A}_M(x; \mathcal{S}_M)$    ▷ Generate Solutions (Eq. 2)
8:              **else**
9:                  $h_{j-1} \leftarrow$ Summarize the responses from agents in round $j-1$
10:                 $y_{i,j}, \ldots, y_{M,j} \leftarrow \mathcal{A}_1(x; h_{j-1}), \ldots, \mathcal{A}_M(x; h_{j-1})$    ▷ Refine Solutions (Eq. 3)
11:              **end if**
12:          **end for**
13:          $\hat{y} \leftarrow$ Majority Vote of $\{y_{1,N}, \cdots, y_{M,N}\}$
14:          $\mathcal{D}_c \leftarrow$ Select samples aligned with $\hat{y}$          ▷ Correctness Selection (Eq. 4)
15:          $\mathcal{D}_{div} \leftarrow$ Select samples for diversity          ▷ Diversity Selection (Eq. 7)
16:          $\hat{S}_1, \cdots, \hat{S}_M \leftarrow \mathcal{A}_{\text{align}}(S_1, P_{ref}), \cdots, \mathcal{A}_{\text{align}}(S_M, P_{ref})$    ▷ Refine Strategies (Eq. 8)
17:      **end for**
18:      $\mathcal{D}_f = \mathcal{D}_f \bigcup(x, \hat{S}_1), \cdots, \bigcup(x, \hat{S}_M)$
19:      $G \leftarrow$ finetune $(G, \mathcal{D}_f)$
20: **end for**

---

## 4 EXPERIMENTS

We evaluate the method on three widely used benchmarks, as detailed in Section 4.1, and compare its performance against eight strong baselines described in Section 4.2, as well as four representative LLMs. The evaluated LLMs include two closed-source models—GPT-4o-mini by OpenAI

| Method | GPT-4o-mini | | | Nova Micro | | |
|---|---|---|---|---|---|---|
| | MATH | GSM8K | GPQA | MATH | GSM8K | GPQA |
| CoT | 67.32 | 91.55 | 39.20 | 67.20 | 91.23 | 39.31 |
| MT-CoT | 67.94 | 92.03 | 39.36 | 68.74 | 92.02 | 39.50 |
| STaR | 68.47 | 91.69 | 40.52 | 68.95 | 92.22 | 40.02 |
| Step-Back Prompting | 65.60 | 90.31 | 32.80 | 66.58 | 90.00 | 32.44 |
| Multi-Agent Debate | 70.57 | 92.91 | 40.36 | 70.34 | 92.44 | 41.23 |
| Self-Reflection | 67.75 | 90.25 | 39.28 | 66.74 | 91.25 | 38.62 |
| Self-Contrast | 62.43 | 90.13 | 37.93 | 63.76 | 90.18 | 36.57 |
| DMAD | 71.54 | 93.27 | 42.11 | 71.02 | 92.45 | 42.94 |
| CMAD (Ours) | **74.52** | **94.42** | **44.29** | **73.87** | **94.12** | **45.15** |

| Method | LLaMA-3-8B | | | Qwen2.5-7B | | |
|---|---|---|---|---|---|---|
| | MATH | GSM8K | GPQA | MATH | GSM8K | GPQA |
| CoT | 25.43 | 76.10 | 27.84 | 70.43 | 90.27 | 35.18 |
| MT-CoT | 27.42 | 77.62 | 27.98 | 72.74 | 90.74 | 36.45 |
| STaR | 30.24 | 79.03 | 28.76 | 75.40 | 96.71 | 35.61 |
| Step-Back Prompting | 24.87 | 75.31 | 24.43 | 69.52 | 88.29 | 33.52 |
| Multi-Agent Debate | 30.82 | 78.56 | 28.96 | 75.52 | 92.03 | 37.15 |
| Self-Reflection | 26.32 | 77.48 | 26.92 | 69.85 | 89.31 | 34.72 |
| Self-Contrast | 27.31 | 76.17 | 23.65 | 68.52 | 88.94 | 34.61 |
| DMAD | 31.24 | 78.42 | 30.37 | 76.80 | 92.46 | 37.86 |
| CMAD (Ours) | **33.30** | **82.26** | **31.87** | **78.26** | **93.86** | **39.40** |

Table 1: Quantitative comparison of the proposed method against eight baseline approaches and four mainstream large language models. Best-performing scores are highlighted in gray. Notably, the model is fine-tuned for a single epoch to achieve these results. Further improvements are expected with additional training epochs as shown in Figure 3.

(Achiam et al., 2023) and Amazon Nova Micro (Intelligence, 2024)—and two publicly available models—LLaMA-3-8B-Instruct (Grattafiori et al., 2024) and Qwen2.5-7B-Instruct (Yang et al., 2024). For training details and further experiments on reasoning models, we report it in A and B correspondingly. We also report alternative finetuning method in C and computation analysis in F.

## 4.1 BENCHMARKS

**MATH** is a widely used benchmark comprising problems from high school mathematics competitions, spanning seven distinct subjects (Hendrycks et al., 2021). Accuracy is measured by comparing model predictions against ground truth answers, and correctness is determined through exact match.

**GSM8K** is a benchmark dataset comprising math word problems that require multi-step reasoning to solve (Cobbe et al., 2021). Each example consists of a problem statement, a corresponding numerical answer, and a step-by-step explanation.

**GPQA** is a challenging dataset consisting of 448 multiple-choice questions meticulously crafted by domain experts in biology, physics, and chemistry (Rein et al., 2024).

## 4.2 BASELINES

We compare our proposed method against several state-of-the-art baselines. For our method, we employ three agents ($M = 3$), and for all debate-based approaches, we conduct two rounds of debate ($N = 2$). The baselines are as follows:

**Chain-of-Thought Prompting (CoT)** (Wei et al., 2022) This approach enables large language models to decompose complex problems by generating intermediate reasoning steps that lead to the final answer, thereby enhancing problem-solving capabilities through explicit step-by-step reasoning.

**Majority Voting with Chain-of-Thought Prompting (MV-CoT)** Wang et al. (2025) This approach is based on CoT with several forward passes. The final result is selected from the most voted answers.

**STaR** (Zelikman et al., 2022) The method operates through an iterative learning loop. Starting with a few example rationales, it generates reasoning paths to answer multiple questions. When answers are incorrect, the system attempts new rationales while knowing the correct answer. The model is then fine-tuned on all successful rationales—those that led to correct answers. Then such process repeats.

**Step-Back Prompting** (Zheng et al., 2023) The method improves reasoning by first prompting the model to abstract the problem to higher-level concepts, then applying these abstractions to solve the problem, effectively separating conceptual understanding from solution execution.

**Multi-Agent Debate** (Du et al., 2023) This framework facilitates multi-agent interaction where agents iteratively critique and refine solutions through structured debates, leveraging diverse perspectives to converge toward more robust answers.

**Self-Reflection** (Madaan et al., 2023) This technique enables models to evaluate and refine their initial outputs by critically analyzing their own reasoning, identifying potential errors or limitations, and generating improved solutions based on this introspection.

**Self-Contrast** (Zhang et al., 2024a)The proposed method generates diverse reasoning paths, identifies their discrepancies, and distills these differences into a structured checklist. The model then reflects on this checklist to iteratively revise each reasoning path, aiming to reach a coherent consensus.

**DMAD** (Liu et al., 2025) This method applies a set of pre-defined reasoning strategies to generate diverse solution paths, encouraging exploration of multiple problem-solving approaches.

### 4.3 QUANTITATIVE RESULTS

We report the performance of the proposed method compared to eight baseline approaches and four mainstream large language models. Notably, the strategy generator is fine-tuned for only a single iteration, i.e., $L = 1$. Results with additional refinement rounds are presented in Section 5. As shown in Table 4.1, the proposed method consistently outperforms all baselines. The average improvement over the second-best method ranges from 1.2% to 9.8%. These results demonstrate that the proposed approach effectively generates feasible strategies to guide LLMs in solving complex questions.

Notably, additional fine-tuning iterations lead to improved performance. For example, as shown in Figure 3, we observe an improvement in accuracy reaching 18% for LLaMA-3-8B on the math datasets. Additionally, as we include more LLM agents, the performance increases, as shown in the left part of Figure 4. Additional debate rounds also benefit the final results.

## 5 DISCUSSIONS AND VISUALIZATIONS

We present additional analyses and visualizations of results on `GPT-4o-mini` and `LLaMA-3-8B`. Specifically, we investigate the following questions: (1) What is the contribution of each component within the proposed framework? (2) How does the method perform with additional rounds of fine-tuning? (3) How does the diversity of solutions evolve across different stages of fine-tuning? (4) How does the threshold for diversity affect the overall performance of the framework? Additional experimental results, including the framework's compatibility with different fine-tuning methods and limitations, are provided in Appendix-A3 and Appendix-A4, respectively.

**Ablation Studies**    We are interested in the variants of CMAD in the following settings:

*CMAD with Pre-defined Strategy*: We adopt three predefined reasoning strategies suggested by prior work to generate fixed strategies for the guidance of solution generation (Liu et al., 2015).

*CMAD w/o Sample Selection*: All generated strategies are used for fine-tuning, regardless of their correctness or diversity.

| Method | GPT-4o-mini | | | LLaMA-3-8B | | |
|---|---|---|---|---|---|---|
| | MATH | GSM8K | GPQA | MATH | GSM8K | GPQA |
| CMAD with Pre-defined Strategies | 70.62 | 92.89 | 42.07 | 32.11 | 78.20 | 29.23 |
| CMAD w/o Sample Selection | 70.23 | 91.39 | 40.34 | 30.62 | 77.62 | 28.63 |
| CMAD w/o Diversity Selection | 71.03 | 91.54 | 41.74 | 30.77 | 78.04 | 30.04 |
| CMAD w/o Correctness Selection | 71.83 | 92.82 | 41.21 | 31.16 | 78.28 | 29.35 |
| CMAD w/o Strategy Refinement | 74.22 | 93.85 | 43.27 | 33.02 | 81.88 | 31.43 |
| CMAD (Ours) | **74.52** | **94.42** | **44.29** | **33.30** | **82.26** | **31.87** |

Table 2: **Ablation Results**: We analyze the contributions of individual components of the proposed method to overall performance.

*CMAD w/o Correctness Selection*: Only diverse strategies are used for fine-tuning, without explicitly matching them to pseudo labels.

*CMAD w/o Diversity Selection*: Strategies aligned with pseudo labels are used for fine-tuning, without explicitly enforcing diversity constraints.

*CMAD w/o Strategy Refinement*: Strategies aligned with pseudo-label diversity are used directly for fine-tuning, without any iterative refinement or revision.

The results are summarized in Table 5. We observe that fixed strategies guided by pre-defined templates yield similar results to previous research (Liu et al., 2025). Our experiments demonstrate that sample selection significantly improves the overall performance of the framework, suggesting that high-quality examples are critical for effective fine-tuning. Specifically, correctness-based selection produces gains by aligning the strategy generator with correct solutions. Although the improvement is modest, incorporating diversity further enhances performance. Moreover, strategy refinement contributes additional improvement by calibrating the initial strategies based on insights derived from alternative solutions.

**Performance with Multiple Iterations of Finetuning**  To verify the effectiveness of multiple iterations of fine-tuning, we report the performance of CMAD over five iterations in Figure 3. CMAD consistently improves the final results as the training iterations increase, with `GPT-4o-mini` varying from 74.52% to 76.80% and `LLaMA-3` from 33.30% to 37.04%. This improvement stems from the diverse samples selected purposefully.

In contrast, fine-tuning examples without considering diversity saturates after one iteration of fine-tuning and even begins to produce worse results. This observation is attributed to overfitting similar solutions, ultimately leading to the collapse of the training process. Additionally, we visualize the results with pre-defined strategy without fine-tuning as one of the baselines for comparison. We observe that overfitting with correct but less diverse samples could lead to worse results than pre-defined strategies, thereby further proving the importance of data quality and selection process.

**Performance with Additional Agents and Rounds of Debate**  We report the results of DMAD with additional debate agents using with GPT-4o-mini in Figure 4.

*Number of Agents:* We analyze the impact of the number of agents in debate. We fix all debate rounds to two, having only one iteration of fine-tuning and applying the summarization trick as usual. We notice the accuracy consistently increases with additional agents, i.e., the number of LLM solvers involved.

*Rounds of Debate:* We analyze the impact of the number of debate rounds in multiagent debate. We fix the number of agents to three as the standard setting. We observe the performance benefits from further debate; however, it may saturate after three rounds.

**Diversity with Multiple Iterations of Finetuning**  While we encourage the strategy generator $G$ to explore diverse strategies through simple prompt-level instructions, a single agent tends to converge toward generating similar—albeit correct—solutions after multiple rounds of fine-tuning. To better

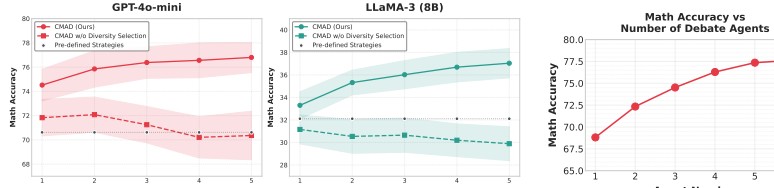

Figure 3: Performance with more iterations. Insufficient diversity in training examples ultimately degrades training effectiveness (dashed red lines), resulting in performance worse than debate methods with predefined strategies (dashed gray lines).

Figure 4: **Left:** Performance on MATH consistently improves as the number of agents involved increases. **Right:** Performance on MATH is enhanced as the number of rounds of underlying debate increases.

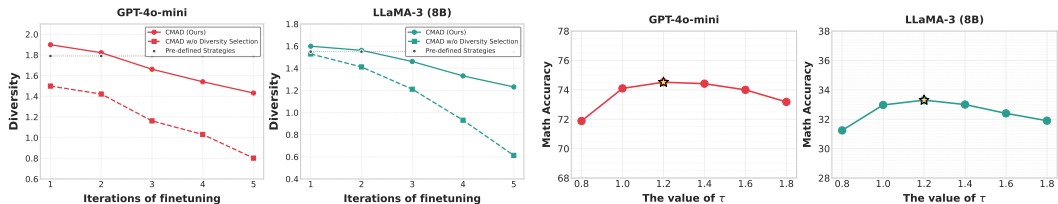

Figure 5: **Left:** Diversity (↑) under the critical thinking framework on the MATH dataset, measured via the uniformity magnitude (Equation 7). Diversity decline is mitigated with diversity-based sampling (solid red), while human-defined strategies remain stable due to the absence of fine-tuning (dashed gray). **Right:** CMAD accuracy across diversity thresholds $\tau$ using MATH, with optimal results near $\tau = 1.2$.

understand this behavior, we report the diversity of generated solutions, quantified by the uniformity metric defined in Equation 7. For more intuitive interpretation, we define the diversity as the absolute value of uniformity in Figure 5. We observe that CMAD maintains diversity within a relatively stable range, in contrast to variants without diversity-based sample selection. For comparison, we also visualize the diversity of solutions generated by human-defined strategies, which serve as an idealized baseline for diversity. Since no fine-tuning is applied in these predefined settings, their diversity remains constant throughout.

**Threshold of Diversity**   Based on prior experiments, insufficient diversity leads to overfitting on semantically similar yet correct samples, as shown in Figure 3. However, setting the diversity threshold $\tau$ too high reduces the pool of eligible training samples, thereby constraining the capacity of the fine-tuned strategy generator. To better understand this trade-off, we analyze the effect of varying $\tau$, as illustrated in Figure 5. When $\tau$ is low, model performance aligns with the baseline lacking diversity-based selection. As $\tau$ increases, CMAD performance improves, peaking at an optimal threshold before degrading due to insufficient training data. Notably, the effective range of $\tau$ increases with more capable LLMs, which naturally generate higher-quality and more diverse strategies.

## 6   CONCLUSION

We introduced Critical Thinking with Multi-Agent Debate (CMAD), a novel framework that stimulates the latent creativity of LLMs by encouraging the generation of diverse and undefined solutions. By employing a strategy generator, the proposed method automatically equips multiple agents with distinct roles and reasoning pathways to collaboratively address and solve complex problems. We further introduce a feedback mechanism grounded in `Correctness` and `Diversity` to ensure the selection of high-quality solutions. These solutions are then used to fine-tune the strategy generator, promoting both creativity and reliability. Notably, CMAD enables autonomous self-improvement through iterative fine-tuning, achieving substantial performance gains without incurring heavy computational costs. We hope this work provides new insights into multi-agent debate, fine-tuning for self-correction, and the broader development of LLMs.

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

## OVERVIEW OF THE APPENDIX

The Appendix is organized as follows:

- Appendix A describes the general experimental setup.
- Appendix B introduces additional results with reasoning models.
- Appendix C presents results with different fine-tuning methods.
- Appendix D details the prompts used for strategy refinement.
- Appendix E discusses the limitations of this work.
- Appendix F reports related compute cost.

## A    EXPERIMENTAL SETUP

For all open-source models, the training is conducted on **p4d.24xlarge**: 8× NVIDIA A100 (40GB) GPUs or **g5.12xlarge**: 4× NVIDIA A10G (24GB) GPUs. For full model fine-tuning of the Qwen1.5-7B and LLaMA-8B models, the training process uses bfloat16 precision with a global batch size of 32 achieved through gradient accumulation steps of 16. We apply a conservative learning rate

| Method | DeepSeek-R1-Distill-Qwen-7B | | DeepSeek-R1-Distill-Llama-8B | |
| --- | --- | --- | --- | --- |
| | MATH | GPQA | MATH | GPQA |
| Without DMAD | 90.27 | 51.32 | 87.36 | 50.48 |
| With DMAD | **92.80** | **51.90** | **91.08** | **51.98** |

Table 3: Quantitative comparison of the proposed method against baseline approaches and mainstream large language models. Best-performing scores are in bold. The CMAD-LoRA variant uses parameter-efficient fine-tuning with approximately 0.1% of trainable parameters compared to full fine-tuning, while still outperforming most baseline methods. All models are fine-tuned for a single epoch to achieve these results.

of 5e-6 with 3% linear warmup followed by cosine decay scheduling to maintain training stability. Regularization techniques include gradient clipping at 1.0, weight decay of 0.01 applied using the AdamW optimizer, and a dropout rate of 0.1 throughout the network. The training runs for 1 epoch with early stopping based on validation loss plateau detection, evaluated every 100 training steps to achieve optimal performance while preventing overfitting on the target domain.

The method also supports LoRA for efficient parameter-efficient fine-tuning. Specifically, we utilized LoRA (Low-Rank Adaptation) with rank $r = 16$, scaling factor $\alpha = 32$, and a conservative dropout rate of 0.05, targeting the attention modules (q_proj, v_proj, k_proj, o_proj). To maintain consistency with full fine-tuning while accounting for LoRA's characteristics, we implemented gradient check-pointing, and used a global batch size of 16 (per-device batch size of 1 with gradient accumulation steps of 16). The training process used the AdamW optimizer with fused implementation where available, a learning rate of 5e-5 (adjusted for the efficient parameterization), 3% linear warmup steps, gradient clipping at 1.0, and no weight decay applied to the LoRA parameters themselves to avoid over-regularizing the low-rank matrices.

## B    PERFORMANCE OF REASONING LLM WITH CMAD

We further report the performance results of more recent reasoning LLMs using CMAD models, including DeepSeek-R1-Distill-Qwen-7B and DeepSeek-R1-Distill-Llama-8B (Guo et al., 2025). The results are shown in Table 3, following the settings of main experiments with three separate agents and two rounds of debate. The improvements are consistent across all metrics when the reasoning process is present.

## C    COMPARISON OF DIFFERENT FINE-TUNING METHODS

The proposed CMAD framework demonstrates exceptional adaptability to parameter-efficient fine-tuning approaches as shown in Table C. When implemented with LoRA, CMAD maintains around 96% of its performance on average across all benchmarks. Specifically, we use Qwen2.5-7B as the univeral strategy generator, but different LLM solver such as GPT-4o-mini, Nova Micro, LLaMA-3-8B.

These results consistently surpass established reasoning baselines irrespective of the underlying parameter optimization technique. This invariance to fine-tuning methodology suggests that CMAD's performance improvements derive from its collaborative reasoning architecture rather than specific parameter updates.

## D    STRATEGY REFINEMENT

The strategy alignment process is a critical component of our Critical Thinking with Multi-Agent Debate (CMAD) framework. This process ensures coherence between the initial high-level strategies generated by the strategy generator and the concrete solutions produced after multiple rounds of debate. In the multi-agent debate process, the initial strategy ($S_i$) guides an agent's reasoning

| Method | Solver: GPT-4o-mini | | | Solver: Nova Micro | | |
|---|---|---|---|---|---|---|
| | MATH | GSM8K | GPQA | MATH | GSM8K | GPQA |
| CoT | 67.32 | 91.55 | 39.20 | 67.20 | 91.23 | 39.31 |
| Step-Back Prompting | 65.60 | 90.31 | 32.80 | 66.58 | 90.00 | 32.44 |
| Multi-Agent Debate | 70.57 | 92.91 | 40.36 | 70.34 | 92.44 | 41.23 |
| Self-Reflection | 67.75 | 90.25 | 39.28 | 66.74 | 91.25 | 38.62 |
| Self-Contrast | 62.43 | 90.13 | 37.93 | 63.76 | 90.18 | 36.57 |
| DMAD | 71.54 | 93.27 | 42.11 | 71.02 | 92.45 | 42.94 |
| CMAD (Ours) | **73.25** | **93.88** | **44.03** | **74.78** | **93.71** | **45.22** |
| CMAD-LoRA (Ours) | 71.84 | 93.35 | 42.67 | 71.25 | 92.81 | 42.41 |

| Method | Solver: LLaMA-3-8B | | | Solver: Qwen2.5-7B | | |
|---|---|---|---|---|---|---|
| | MATH | GSM8K | GPQA | MATH | GSM8K | GPQA |
| CoT | 25.43 | 76.10 | 27.84 | 70.43 | 90.27 | 35.18 |
| Step-Back Prompting | 24.87 | 75.31 | 24.43 | 69.52 | 88.29 | 33.52 |
| Multi-Agent Debate | 30.82 | 78.56 | 28.96 | 75.52 | 92.03 | 37.15 |
| Self-Reflection | 26.32 | 77.48 | 26.92 | 69.85 | 89.31 | 34.72 |
| Self-Contrast | 27.31 | 76.17 | 23.65 | 68.52 | 88.94 | 34.61 |
| DMAD | 31.24 | 78.42 | 30.37 | 76.80 | 92.46 | 37.86 |
| CMAD (Ours) | **35.30** | **82.72** | **32.54** | **78.26** | **93.86** | **39.40** |
| CMAD-LoRA (Ours) | 32.91 | 80.94 | 30.82 | 76.93 | 92.42 | 37.88 |

Table 4: Quantitative comparison of the proposed method against eight baseline approaches and four mainstream large language models. Best-performing scores are highlighted in bold. The CMAD-LoRA variant uses Qwen2.5-7B as the strategy generator, while using different LLMs as solvers following the strategies generated.

approach. However, as agents engage in critique and refinement throughout debate rounds, the final solution ($y_{i,N}$) may deviate from or extend beyond the initial strategic guidance. This misalignment can reduce the effectiveness of fine-tuning, as the strategy generator would learn to produce strategies that don't accurately reflect successful reasoning paths. To address this challenge, we introduce a strategy alignment agent $A_{\text{align}}$ that examines both the initial strategy $S_i$ and the final solution $y_{i,N}$ (especially when this solution matches the pseudo-ground truth $\hat{y}$). The alignment agent then performs one of two actions:

1. Validates the initial strategy if it aligns well with the successful solution

2. Refines the strategy to better capture the actual reasoning path that led to the correct answer

The refined strategies $\hat{S}_i$ are then paired with the original problems to create high-quality fine-tuning data for the strategy generator.

### D.1 STRATEGY ALIGNMENT ALGORITHM

### D.2 ALIGNMENT PROMPT TEMPLATE DESIGN

The design of the alignment prompt template ($P_{\text{ref}}$) is critical for ensuring consistent and meaningful strategy refinement. This prompt instructs the strategy alignment agent ($A_{\text{align}}$) to perform a rigorous analysis of the relationship between the initial strategy and the successful solution pathway. The template is structured to facilitate both qualitative assessment and concrete refinement actions.

---

**Algorithm 2** Strategy Alignment Process

---

**Require:** A set of initial strategies $S_i|i = 1, 2, ..., M$; Final solutions after $N$ rounds of debate
  $y_{i,N}|i = 1, 2, ..., M$; Pseudo label $\hat{y}$ from majority vote; Strategy alignment agent $A_{\text{align}}$; Alignment prompt template $P_{\text{ref}}$

 1: Initialize aligned strategy set $\hat{S}i = \emptyset$
 2: **for** $i = 1 \rightarrow M$ **do**
 3:    **if** $yi, N = \hat{y}$ **then**
 4:       // Construct the alignment prompt with specific instruction
 5:       $P_{\text{align}} \leftarrow \text{Format}(P_{\text{ref}}, \text{problem} = x, \text{strategy} = S_i, \text{solution} = y_{i,N})$
 6:       // Send to the alignment agent to evaluate and potentially refine the strategy
 7:       $\text{strategyEvaluation} \leftarrow A_{\text{align}}(P_{\text{align}})$
 8:       **if** strategyEvaluation indicates "aligned" **then**
 9:         $\hat{S}_i \leftarrow S_i$ // Keep original strategy if already well-aligned
10:       **else**
11:         // Extract the refined strategy from the agent's response
12:         $\hat{S}_i \leftarrow \text{ExtractRefinedStrategy}(\text{strategyEvaluation})$
13:       **end if**
14:       // Add to the set of aligned strategies paired with the original problem
15:       Add $(x, \hat{S}_i)$ to fine-tuning dataset $D_f$
16:    **end if**
17: **end for**
18: **return** $\hat{S}_i|i = 1, 2, ..., M$ // Return the set of aligned strategies

---

**Prompt Structure and Components** The prompt template comprises several key components designed to guide the alignment agent's analysis:

1. **Context Specification**: Presentation of the original problem, initial strategy, and final solution

2. **Analytical Framework**: Specific evaluation criteria for assessing strategy-solution alignment

3. **Refinement Guidelines**: Structured approach for strategy modification when misalignment is detected

4. **Response Format**: Standardized output format to ensure consistency across evaluations

**Alignment Prompt Template** The complete alignment prompt template is formalized as follows:

```
STRATEGY ALIGNMENT EVALUATION
You are conducting a formal evaluation of the alignment between a proposed
problem solving strategy and the actual solution trajectory that led to
a correct answer.

Your objective is to determine whether refinement of the original strategy is
necessary.

PROBLEM STATEMENT:
{problem}
ORIGINAL STRATEGY:
{strategy}
SUCCESSFUL SOLUTION:
{solution}
EVALUATION INSTRUCTIONS:

ANALYTICAL ASSESSMENT:
a. Identify the key reasoning steps in the solution
b. Compare these reasoning steps with the guidance provided in the original strategy
c. Determine whether there are significant deviations in:
```

```
Conceptual approach
Reasoning sequence
Computational methodologies
Special case handling
d. Evaluate whether the solution introduces novel or more effective approaches
not captured in the original strategy

ALIGNMENT DETERMINATION:
The original strategy and solution are considered ALIGNED when:

The solution follows the general reasoning framework outlined in the strategy
Any deviations are minor and do not alter the core approach
The strategy successfully captures the essential characteristics of the
solution path

The strategy and solution are considered MISALIGNED when:

The solution employs significantly different reasoning approaches
The solution introduces critical steps not anticipated in the strategy
The strategy emphasizes elements that proved unnecessary in the actual
solution. The solution path represents a more efficient or
generalizable approach

STRATEGY REFINEMENT (if necessary):
If misalignment is detected, formulate a refined strategy that:

Accurately captures the successful reasoning pattern demonstrated in
the solution
Preserves general applicability to similar problem types
Incorporates novel insights or approaches revealed in the solution
Provides clear and actionable guidance for solving similar problems

RESPONSE FORMAT:
[Alignment Analysis]
A comprehensive analysis comparing the strategy and solution, identifying
specific points of alignment or misalignment with explicit references to
both components.
[Alignment Status]
ALIGNED or MISALIGNED (select exactly one)
[Refined Strategy]
If status is ALIGNED: "Original strategy retained"
If status is MISALIGNED: Provide a complete reformulation of the
strategy that accurately reflects the reasoning path demonstrated
in the successful solution.
```

This structured prompt design ensures that the alignment agent can systematically evaluate and refine strategies, thereby contributing to the continuous improvement of the strategy generator. By formalizing the criteria for alignment and the process for refinement, we establish a consistent mechanism for capturing successful reasoning pathways and incorporating them into the model's strategic repertoire.

The alignment evaluation process produces three possible outcomes:

1. **Confirmed Alignment**: The original strategy is validated as effective and retained without modification.

2. **Minor Refinement**: The core approach is sound, but the strategy is enhanced with additional details or clarifications identified in the solution.

3. **Major Reformulation**: A significantly different strategy is formulated to capture novel reasoning approaches demonstrated in the solution.

# E  LIMITATIONS

While our Critical Thinking with Multi-Agent Debate framework demonstrates significant improvements in reasoning performance across various models and tasks, several limitations warrant consideration. Although CMAD offers computational advantages over traditional fine-tuning approaches by focusing on strategy generation rather than comprehensive model retraining, it still incurs non-trivial computational costs when deploying multiple agents for each problem, especially with larger models. This resource requirement, while more efficient than full model fine-tuning, may still constrain applications in resource-limited environments. The strategy generator's performance is inherently bounded by the reasoning capabilities of the underlying LLMs, potentially limiting its effectiveness with less capable models.

# F  COMPUTATION ANALYSIS

**Cost for Finetuning** With default settings, one iteration of data generation, collection, and fine-tuning over qualified data pairs from 1000 problems from MATH takes 12-20 hours on a p4d.24xlarge instance. The training process could be further accelerated through parallelization of the generation phase.

**Cost for Inference** We report a computational analysis on the MATH dataset using GPT-4o-mini, with token costs detailed in Table F. The total tokens represent the combined prompt and completion tokens. Our inference settings align with the main experiments. For inference time, processing 200 problems took between 2-3 hours using eight GPUs, with variation depending on model size.

| Methods | Tokens | Cost ($) | Accuracy |
|---|---|---|---|
| CoT | 1,206,125 | 0.38 | 67.32 |
| Self-Reflection | 3,323,230 | 1.06 | 67.75 |
| Self-Contrast | **7,371,764** | 2.35 | 62.43 |
| Multi-Agent Debate | 4,531,212 | 1.44 | 70.57 |
| DMAD | 7,280,435 | 2.42 | 71.54 |
| CMAD (Ours) | 7,210,359 | 2.30 | **74.52** |

Table 5: Tokens and cost overhead of different methods on MATH for inference time.

