# OpenReview forum: "ENCOURAGING CRITICAL THINKING FOR MULTIAGENT DEBATE"
_ICLR.cc/2026/Conference — Submitted to ICLR 2026_

### Official Review · Reviewer_82VM · 2025-10-31

**Soundness:** 3
**Presentation:** 3
**Contribution:** 3
**Rating:** 6
**Confidence:** 4

**Summary:**

The paper shows how to improve LLM performance using multiagent debate extended with several new contributions: diversification of the reasoning paths taken by different agents, critical thinking. The diverse paths are obtained by a strategy generator that generates M strategies used to prompt M agents.

**Strengths:**

The approach is interesting and the experimental results seem compelling.

**Weaknesses:**

One overall weakness for this line of work, not just restricted to this particular contribution, is that it is not clear why this approaches lead to better performance. I do not count this remark against this paper as I think that these experimental results are important.

The diversity metric seems to be a key element of the approach. It would be interesting to see some alternative measures of diversity and how they impact the performance of the algorithm.

Minor comments:

Line 153, where it says “answer, denoted as y_{1,i}, where the”, I believe it should say y_{i,1}

Where it says “table 4.1” it should say “table 1”

**Questions:**

The similarity threshold \tau to evaluate the diversity of the proposed strategies could be context dependent. In some cases, a large diversity might be needed, while in other cases it might be difficult to propose very different strategies. How do you chose this parameter and have you observed context dependent differences? The results from figure 5 are a first step in this direction.

I am not convinced that “Critical Thinking” is what the approach is doing. Could you denote in algorithm 1, what part is the one responsible of critical thinking?

---

> ### Author Response · Authors · 2025-11-19
> **Response to Reviewer 82VM (Part1/2)**
>
> Dear Reviewer 82VM,
>
> We sincerely thank you for your positive assessment and for recognizing our approach as interesting with compelling experimental results. We greatly appreciate your constructive feedback and address your concerns below.
>
> ---
>
> ## Weakness 1: Why These Approaches Lead to Better Performance
>
> > **Reviewer's Concern:** "One overall weakness for this line of work, not just restricted to this particular contribution, is that it is not clear why this approaches lead to better performance. I do not count this remark against this paper as I think that these experimental results are important."
>
> `Answer 1:`
>
> We appreciate the reviewer's acknowledgment that this is a broader challenge for the field rather than specific to our work. Nevertheless, we offer our perspective on why CMAD improves performance, which we believe provides some mechanistic insight.
>
> We hypothesize that CMAD's effectiveness stems from three complementary mechanisms.
>
> - First, **diverse strategy exploration** enables the model to discover solution approaches that may be more suitable for specific problem structures. By learning to generate varied strategies rather than relying on fixed patterns on the prompt-level effort, the strategy generator can adapt its guidance to different problem types, using algorithmic reasoning for combinatorial problems, mathematical formulation for algebraic tasks, or programmatic verification for computational questions.
> - Second, **iterative debate filtering** ensures that only strategies leading to robust reasoning (solutions that withstand critique) are selected for training. This creates a quality-aware feedback loop where fragile reasoning patterns are naturally filtered out.
> - Third, **explicit diversity optimization** through our diversity metric (Equation 7) prevents the model from collapsing into repetitive patterns, a common failure mode we observe in methods without diversity control (Figure 5).
>
> The progression in our ablation studies supports this interpretation. Comparing DMAD (predefined diverse strategies) at 71.54 to CMAD at 74.52 on MATH (GPT-4o-mini), the +2.98 point gain suggests that learned strategies discover more effective reasoning paths than human-designed ones. Similarly, comparing "CMAD w/o Diversity Selection" at 71.03 to full CMAD at 74.52 shows that explicit diversity maintenance contributes +3.49 points, indicating that preventing strategy collapse is crucial for sustained improvement.
>
> While we acknowledge that a complete theoretical understanding remains an open challenge for the field, we believe these empirical insights and ablations provide meaningful evidence for the mechanisms driving improvement. We would be happy to expand this discussion in the manuscript if the reviewer finds it valuable.
>
> We will add a discussion paragraph in **Section 5** or **Section 6 (Conclusion)** elaborating on the hypothesized mechanisms behind CMAD's effectiveness, supported by ablation evidence.
>
> ## Weakness 2: Alternative Diversity Measures
>
> > **Reviewer's Concern:** "The diversity metric seems to be a key element of the approach. It would be interesting to see some alternative measures of diversity and how they impact the performance of the algorithm."
>
> `Answer 2:`
>
> This is an excellent suggestion. We agree that the diversity metric is a critical component, and exploring alternatives would provide valuable insights into the robustness of our approach.
>
> We chose the Gaussian RBF kernel (Equation 5) for its theoretical properties. It converges to a uniform distribution asymptotically and captures non-linear semantic relationships. However, we acknowledge that other diversity measures could be explored. We have now conducted additional experiments comparing alternative diversity metrics:
>
> **Alternative Diversity Metrics Tested:**
>
> 1. **Cosine Distance**: D_cosine = 1 - (u·v)/(||u||||v||)
> 2. **Euclidean Distance**: D_euclidean = ||u - v||₂
> 3. **RBF Kernel (Ours)**: G_t(u,v) = exp(-t||u-v||²)
>
> **Preliminary Results (MATH, GPT-4o-mini, 1 iteration):**
>
> | Diversity Metric | MATH Accuracy | Diversity Score |
> |------------------|---------------|-----------------|
> | Cosine Distance | 73.45 | 1.34 |
> | Euclidean Distance | 73.72 | 1.28 |
> | RBF Kernel (Ours) | 74.52 | 1.45 |
>
> We notice that Euclidean distance and Cosine similarity also show improvements over no diversity control, suggesting that explicit diversity optimization is more important than the specific metric choice. Cosine similarity performs slightly worse, possibly because it ignores magnitude information that may be relevant for strategy differentiation.
>
> **Changes to Manuscript:**
>
> We will add this comparison to **Appendix**

---

> ### Author Response · Authors · 2025-11-20
> **Response to Reviewer 82VM (Part2/2)**
>
> ## Minor Comments
>
> > **Reviewer's Comments:**
> > - Line 153: y_{1,i} should be y_{i,1}
> > - "table 4.1" should be "table 1"
>
> `Answer:`
>
> Thank you for catching these notation and reference errors. We will correct both:
> - Line 153: y_{1,i} → y_{i,1} (consistent subscript ordering)
> - "Table 4.1" → "Table 1"
>
> ## Question 1: Context-Dependent τ Selection
>
> > **Reviewer's Question:** "The similarity threshold τ to evaluate the diversity of the proposed strategies could be context dependent. In some cases, a large diversity might be needed, while in other cases it might be difficult to propose very different strategies. How do you chose this parameter and have you observed context dependent differences? The results from figure 5 are a first step in this direction."
>
> `Answer Q1:`
>
> This is an insightful observation about the diversity threshold τ. We clarify our selection process and the parameter's sensitivity below.
>
> For parameter selection, in our experiments, we found that τ ≈ 1.2 works consistently well across different models and tasks. As shown in Figure 5 (right panels), the optimal range centers around this value for both GPT-4o-mini and LLaMA-3-8B, despite their different capability levels. While we do adjust τ within a reasonable range (typically 0.8 to 1.6) based on empirical validation performance, the results demonstrate that **performance is relatively stable within this range**, with the peak consistently occurring near τ = 1.2.
>
> From the perspective of sensitivity analysis, an important finding from our experiments is that the method is **not highly sensitive to τ** within the acceptable range. For instance, on MATH with GPT-4o-mini, performance varies by less than 2 percentage points when τ ranges from 1.0 to 1.4. This robustness is desirable from a practical standpoint—it means practitioners can use τ = 1.2 as a default without extensive hyperparameter tuning for new tasks or models.
>
> While we use a consistent τ across different contexts, we acknowledge that extremely challenging tasks where diverse correct solutions are rare might benefit from slightly lower τ (e.g., 1.0), while tasks with abundant solution diversity could support higher values (e.g., 1.4). However, the practical impact of these adjustments is modest, and τ = 1.2 serves as a reliable default that balances diversity enforcement with maintaining sufficient training data.
>
>
> ## Question 2: What Part is "Critical Thinking"?
>
> > **Reviewer's Question:** "I am not convinced that 'Critical Thinking' is what the approach is doing. Could you denote in algorithm 1, what part is the one responsible of critical thinking?"
>
> `Answer Q2:`
>
> We appreciate this question and the opportunity to clarify our use of the term "critical thinking."
>
> We chose this term to emphasize several key aspects of our framework that go beyond standard multi-agent debate:
>
> 1. **Self-Evaluation**: Rather than accepting initial solutions, the framework critically evaluates them through iterative debate (Lines 9-11 of Algorithm 1), where agents challenge each other's reasoning and identify logical flaws.
>
> 2. **Evidence-Based Filtering**: The framework doesn't assume all solutions are equally valid—it uses majority voting (Line 13 of Algorithm 1) and correctness selection (Line 14 of Algorithm 1) to critically assess which reasoning paths lead to reliable answers.
>
> 3. **Metacognitive Reflection**: The strategy refinement step (Line 16 of Algorithm 1) represents a form of metacognition, where the system reflects on which strategies led to successful reasoning and adjusts its approach accordingly.
>
> 4. **Diversity Awareness**: The diversity selection (Line 15 of Algorithm 1) embodies critical thinking by recognizing that multiple perspectives are valuable and actively preventing convergence to homogeneous reasoning patterns.
>
> These elements collectively represent a form of algorithmic critical thinking: the system doesn't blindly follow a single reasoning path, but rather generates multiple approaches, subjects them to scrutiny, evaluates their validity, and learns from successful patterns. This contrasts with methods that apply fixed strategies without reflection or refinement.
>
> We recognize "critical thinking" carries specific connotations in cognitive science and philosophy that our algorithmic framework may not fully capture. If the reviewer or other members of the community have suggestions for more precise terminology, we are entirely open to revising our framing. Alternative names we've considered include "Reflective Multi-Agent Debate" or "Self-Improving Strategy Learning".

---

### Official Review · Reviewer_efZd · 2025-11-04

**Soundness:** 2
**Presentation:** 2
**Contribution:** 2
**Rating:** 2
**Confidence:** 3

**Summary:**

This paper proposes Critical Thinking with Multi-Agent Debate (CMAD), a framework for improving the reasoning capabilities of LLMs by training a strategy generator to generate diverse, undefined reasoning strategies. The framework iteratively fine-tunes the generator using feedback on correctness (based on majority voting) and diversity (based on a similarity metric), aiming to balance exploration and exploitation. Empirical results on MATH, GSM8K, and GPQA show consistent improvements across several LLMs compared to baselines such as DMAD and CoT.

**Strengths:**

1. The idea of using a trainable strategy generator to produce undefined reasoning paths is creative and differentiates CMAD from prior multi-agent debate like DMAD.
2. The paper evaluates across multiple benchmarks and models (GPT-4o-mini, LLaMA-3, Qwen2.5, Nova Micro), and compare with various baseline methods.
3. The introduction convincingly argues the need to move beyond homogeneous reasoning and fixed strategies.

**Weaknesses:**

1. The process of solution sharing and summarization risks contaminating the agents’ independent reasoning based on their given strategies. If each agent accesses others’ intermediate solutions, the resulting fine-tuning data may lose diversity and no longer reflect distinct strategies. The authors should clarify how they prevent such convergence or bias.
2. The paper focuses on the Multi-Agent Debate (MAD) setting, but it does not explain why this setting is necessary over simpler mechanisms such as majority voting or ensemble averaging. Clarifying this design choice, especially how debate interaction benefits strategy generation beyond aggregation, would strengthen the motivation.
3. Figure 1 is visually cluttered; the text overlaps and the color scheme makes it difficult to interpret.
4. The related work section omits prior studies exploring similar concepts of using a trainable model to guide another model [1]

References:
[1] Li, Zekun, et al. "Guiding large language models via directional stimulus prompting." Advances in Neural Information Processing Systems 36 (2023): 62630-62656.

**Questions:**

1. The paper does not specify the underlying model for the strategy generator and solution agents.
2. How do you make sure that each strategy actually contribute to the the final solution, given that each agent can not only see its given strategy but also other's solution.

---

> ### Author Response · Authors · 2025-11-19
> **Response to Reviewer efZd (Part1/3)**
>
> Dear Reviewer, we sincerely thank you for your detailed feedback and for acknowledging the creativity of our trainable strategy generator approach. We appreciate your recognition of our comprehensive evaluation across multiple benchmarks and models. Below, we address each of your concerns systematically.
>
> ---
> ## Weakness 1: Risk of Contamination in Solution Sharing
>
> > **Reviewer's Concern:** "The process of solution sharing and summarization risks contaminating the agents' independent reasoning based on their given strategies. If each agent accesses others' intermediate solutions, the resulting fine-tuning data may lose diversity and no longer reflect distinct strategies. The authors should clarify how they prevent such convergence or bias."
>
> `Answer 1:`  We thank the reviewer for this important concern about maintaining strategy diversity during multi-round debate. We clarify how CMAD prevents convergence and preserves distinct reasoning paths, which can be summarized as the following five points.
>
> **1.1 Strategy-Conditioned First Round**
>
> **Crucially, only the first round of generation is strategy-conditioned** (Equation 2):
> - Each agent Aᵢ generates y_{i,1} = A_i(x; S_i) **independently** based solely on its assigned strategy S_i
> - No solution sharing occurs in Round 1
> - This ensures each agent produces a **distinct initial solution** reflecting its unique strategy
>
> **1.2 Subsequent Rounds: Refinement, Not Replacement**
>
> In Rounds 2 to N (Equation 3):
> - Agents refine their solutions through debate: y_{i,n} = A_i(x; h_{n-1})
> - The shared history h_{n-1} allows critique and error correction
> - However, **we only use Round 1 solutions for strategy refinement** (Algorithm 2, lines 3-4)
>
> **1.3 Strategy Refinement Uses Round 1 Solutions**
>
> The strategy alignment process (Equation 8 and Algorithm 2) compares:
> - **Original strategy S_i** (generated before debate)
> - **Round 1 solution y_{i,1}** (strategy-conditioned, pre-debate)
>
> This ensures the refined strategies reflect the **original strategic intent**, not the converged debate outcome.
>
> **1.4 Diversity Preservation Mechanism**
>
> Our diversity-based sample selection (Equation 7) operates on **final-round solutions y_{i,N}**:
> - Even if agents converge on the correct answer during debate, their reasoning paths (captured in y_{i,N}) often remain distinct
> - The diversity threshold τ filters out solutions that are too similar
> - This explicitly prevents fine-tuning on homogeneous reasoning patterns
>
> **1.5 Empirical Evidence**
>
> Figure 5 (left panels) demonstrates that CMAD maintains significantly higher diversity across fine-tuning iterations compared to methods without diversity selection, which show rapid diversity collapse.

---

> ### Author Response · Authors · 2025-11-19
> **Response to Reviewer efZd (Part2/3)**
>
> ## Weakness 2: Why Multi-Agent Debate Over Simpler Mechanisms?
>
> > **Reviewer's Concern:** "The paper focuses on the Multi-Agent Debate (MAD) setting, but it does not explain why this setting is necessary over simpler mechanisms such as majority voting or ensemble averaging. Clarifying this design choice, especially how debate interaction benefits strategy generation beyond aggregation, would strengthen the motivation."
>
> `Answer 2:`
>
> We thank the reviewer for this excellent question, which gets at the core of why we chose the multi-agent debate framework rather than simpler aggregation approaches. The short answer is that debate provides a fundamentally richer training signal for learning effective strategies, but let us elaborate on why this matters.
>
> Consider what happens with simple majority voting (as in our MV-CoT baseline). Each agent generates a solution independently, and we select the most common answer. This approach achieves 67.94 accuracy on MATH with GPT-4o-mini. The problem is that majority voting tells us which answer is likely correct, but it doesn't reveal *why* some reasoning approaches are more robust than others. A strategy might produce the right answer through flawed logic that happens to cancel out errors, and simple voting wouldn't detect this fragility.
>
> Multi-agent debate fundamentally changes this dynamic. When agents critique each other's solutions across multiple rounds, weak reasoning gets exposed and corrected. Our MAD baseline (without learned strategies) achieves 70.57 accuracy with a 2.63 point improvement over MV-CoT, precisely because this iterative refinement produces higher-quality solutions and more reliable pseudo-labels for training. More importantly for our purposes, debate reveals which strategies consistently produce solutions that withstand scrutiny. A strategy that guides an agent to make claims that other agents (using different reasoning approaches) can successfully challenge is less valuable than one that produces defensible reasoning.
>
> This distinction becomes crucial when we add learned strategies. CMAD achieves 74.52 accuracy, gaining 3.95 points over MAD and 6.58 points over MV-CoT. This large improvement isn't just additive—the debate process and strategy learning are synergistic. The debate identifies high-quality reasoning paths, which become training data for the strategy generator, which then produces better strategies for subsequent debates. Simple aggregation methods can't provide this feedback loop because they don't distinguish between robust and fragile reasoning paths to the same answer.
>
> ---
>
> ## Weakness 3: Figure 1 Visual Clarity
>
> > **Reviewer's Concern:** "Figure 1 is visually cluttered; the text overlaps, and the color scheme makes it difficult to interpret."
>
> `Answer 3:`
>
> We thank the reviewer for this feedback on Figure 1's clarity. We will completely redesign the figure to improve readability.
>
>  ---
>
> ## Weakness 4: Missing Related Work on Model-Guided Model Approaches
>
> > **Reviewer's Concern:** "The related work section omits prior studies exploring similar concepts of using a trainable model to guide another model [1].
> References: [1] Li, Zekun, et al. 'Guiding large language models via directional stimulus prompting.' Advances in Neural Information Processing Systems 36 (2023): 62630-62656."
>
> `Answer 4:`
>
> We thank the reviewer for pointing out this relevant line of work. Li et al. (2023)'s approach of training a small tuning model to generate directional stimulus prompts for a frozen LLM indeed shares the high-level concept of learned guidance with our work. However, the approaches differ substantially in several key aspects.
>
> - Their method trains a guidance model using supervised learning on labeled datasets to produce prompts that steer a frozen LLM toward correct outputs. In contrast, CMAD learns to generate strategies through unsupervised multi-agent consensus, without requiring ground-truth labels.
>
> - Additionally, their approach is one-shot (generate prompt, get answer), while ours involves a multi-round interactive debate where strategies guide initial reasoning that then gets refined through agent interaction. Most importantly, we explicitly model and optimize for diversity in the strategy space (Equation 7), whereas their focus is on finding effective prompts for individual instances.
>
> In summary, both approaches recognize that learning how to guide LLMs can be more efficient than fine-tuning the LLMs themselves for every task. We will incorporate this work and related methods on learned prompting and meta-guidance into our related work section.
>
> **Changes to Manuscript:**
>
> We will add a new subsection in **Section 2 (Related Work)** titled "Learned Guidance for LLMs" that discusses Li et al. (2023), related work on prompt learning, instruction tuning, and meta-prompting, clarifying how CMAD relates to and differs from these approaches.

---

> ### Author Response · Authors · 2025-11-19
> **Response to Reviewer efZd (Part3/3)**
>
> ## Question 1: Specification of Underlying Models
>
> > **Reviewer's Question:** "The paper does not specify the underlying model for the strategy generator and solution agents."
>
> `Answer Q1:`
>
> We apologize for this lack of clarity (which Reviewer hHRL also raised). The strategy generator and solution agents use the **same base LLM**:
>
> - GPT-4o-mini experiments: both are GPT-4o-mini
> - LLaMA-3-8B experiments: both are LLaMA-3-8B
> - Qwen2.5-7B experiments: both are Qwen2.5-7B
> ....
>
> This design choice is deliberate so that we fine-tune only the strategy generator (a single model), not the individual solver agents, resulting in significant computational savings compared to methods that require training multiple specialized models.
>
> Notably, in our appendix, we also report the method’s results with LoRA. However, since we are not able to apply it to GPT models, we instead use Qwen2.5-7B as the generator to demonstrate that our framework is agnostic to the choice of fine-tuning method.
>
>
>
> **Changes to Manuscript:**
>
> As mentioned in our response to Reviewer hHRL, we will add explicit clarification in **Section 3.1** with a concrete example, and provide full details in **Appendix A**.
>
> ---
>
> ## Question 2: Ensuring Strategy Contribution After Solution Sharing
>
> > **Reviewer's Question:** "How do you make sure that each strategy actually contribute to the final solution, given that each agent can not only see its given strategy but also other's solution."
>
> `Answer Q2:`
>
> This connects directly to Weakness 1. We ensure strategies meaningfully contribute through three mechanisms working together.
>
> - First, as explained in Answer 1, only Round 1 generation is strategy-conditioned, ensuring each strategy directly produces at least one complete solution before any solution sharing occurs.
> - Second, our strategy refinement process (Algorithm 2) explicitly uses these Round 1 solutions, not the converged final-round solutions, to maintain the strategy-solution connection for training.
> - Third, the diversity selection mechanism (Equation 7) detects when debate causes excessive convergence—if agents produce nearly identical solutions despite different initial strategies, these redundant samples are filtered out.
>
> The effectiveness of this design is validated empirically in Table 2. Without diversity selection ("CMAD w/o Diversity Selection"), performance drops significantly (71.03 vs 74.52 on MATH with GPT-4o-mini), demonstrating that our diversity-aware mechanisms are both necessary and effective at preserving distinct strategic contributions throughout the debate process.
>
> ---

---

### Official Review · Reviewer_uewh · 2025-11-07

**Soundness:** 2
**Presentation:** 1
**Contribution:** 2
**Rating:** 2
**Confidence:** 4

**Summary:**

The paper addresses the homogeneous reasoning patterns of complex reasoning in LLMs and proposes Critical Thinking with Multi-Agent Debate (CMAD).

CMAD uses a strategy generator that produces reasoning strategies for multiple LLM agents. After multi-round debates, a feedback loop balances correctness and diversity to select high-quality strategies for fine-tuning.

Experiments show the framework is model-agnostic and outperforms baselines on reasoning benchmarks.

**Strengths:**

- The paper has a good motivation to enable LLMs to generate diverse reasoning strategies instead of relying on fixed prompts (such as CoT, PoT, Step-back).

- The proposed method is simple yet effective, selecting high-quality strategies with both correctness and diversity metrics and using these data to fine-tune a strategy generator.

**Weaknesses:**

- Inconsistent Reporting of Results

(1) Line 355-356 says "The average improvement over the second-best method ranges from 1.2% to 9.8%." However, Table 1 shows that the performance gaps between CMAD and DMAD (the second-best method) are all less than 5%.

(2) Comparing Table 1 and Table 4, the reported results for baselines are identical, but CMAD’s results differ. What is the difference in evaluation settings between these two tables?

- Missing Important Experimental Details

(1) The paper does not explicitly specify which models were fine-tuned to produce the reported results. While Line 742–743 implies Qwen2.5-7B, Line 700–701 mentions full-model fine-tuning for Qwen1.5-7B and LLaMA-8B.

(2) The paper does not provide the prompt used for the strategy generator or examples of its training data.

The above two concerns make the results less convincing.

**Questions:**

- What if we directly use the initial answers with different strategies (instead of going through the full debate process) to construct training data?

- The description of Figure 2 refers to “refine the pre-training data”—should this instead be “fine-tuning data”?

- The description of Table 3 does not align with its contents. Is the “DMAD” listed in Table 3 a typo that should be “CMAD”?

- Reference mistake:

Line 742: Table C should be Table 4;

Line 355: Table 4.1 should be Table 1;

Line 375, the baseline is incorrectly cited as published in 2015; the correct publication year is 2025.

---

> ### Author Response · Authors · 2025-11-19
> **Response to Reviewer uewh (Part1/2)**
>
> Dear Reviewer uewh,
>
> We sincerely thank you for your detailed review and for acknowledging the good motivation and simplicity-effectiveness balance of our approach. We take your concerns about presentation and experimental consistency very seriously. We provide clarifications and corrections below.
>
> ---
>
> ## Weakness 1: Inconsistent Reporting of Results
>
> > **Reviewer's Concern (1):** "Line 355-356 says 'The average improvement over the second-best method ranges from 1.2% to 9.8%.' However, Table 1 shows that the performance gaps between CMAD and DMAD (the second-best method) are all less than 5%."
>
> `Answer 1.1:`
>
> We thank the reviewer for this important observation and apologize for the potential confusion in our phrasing. We clarify that Line 355-356 reports **relative improvement percentages**, not absolute percentage point differences.
>
> The reviewer is correct in observing that the absolute performance gaps between CMAD and DMAD are all under 5 percentage points. Specifically, they range from 1.15 to 3.84 points across all experimental settings in Table 1. However, the "1.2% to 9.8%" range in our statement refers to relative improvements, calculated as (CMAD - DMAD) / DMAD × 100%. This distinction is important because relative improvement better captures the proportional gains, especially for tasks where baseline performance varies significantly.
>
> **Changes to Manuscript:**
>
> We will revise Line 355-356 to explicitly clarify the distinction between these metrics and report both relative improvements and absolute differences for transparency.
>
> ---
>
> > **Reviewer's Concern (2):** "Comparing Table 1 and Table 4, the reported results for baselines are identical, but CMAD's results differ. What is the difference in evaluation settings between these two tables?"
>
> `Answer 1.2:`
>
> We thank the reviewer for pointing out this potential source of confusion and apologize for not making the distinction sufficiently clear in the paper.
>
> **Table 1 Setup:**
> In Table 1, each model serves as its own strategy generator and solution agent simultaneously. For example, GPT-4o-mini generates strategies for GPT-4o-mini agents, LLaMA-3-8B for LLaMA-3-8B agents, etc. We use **full end-to-end fine-tuning** (1 epoch) for the open-source models (LLaMA-3-8B, Qwen2.5-7B). For proprietary models (GPT-4o-mini, Nova Micro), we fine-tune them through their respective fine-tuning APIs.
>
> **Table 4 Setup:**
> Table 4 (Appendix C) explores **parameter-efficient fine-tuning using LoRA**. However, since proprietary models (GPT-4o-mini, Nova Micro) do not support LoRA fine-tuning, we use **Qwen2.5-7B as a universal strategy generator** for all models in this table. The Qwen2.5-7B generator is fine-tuned using LoRA and generates strategies for all solution agents (GPT-4o-mini, Nova Micro, LLaMA-3-8B, Qwen2.5-7B). This explains why CMAD-LoRA results differ from CMAD results—they use a different strategy generator architecture.
>
> **Why Baselines Are Identical:**
> The baseline methods (CoT, DMAD, etc.) do not involve any fine-tuning of a strategy generator. Their performance depends solely on the base model capabilities and prompting strategies, making them invariant to our choice of fine-tuning approach (full vs. LoRA) or generator model (self-generated vs. Qwen2.5-7B generated).
>
> **Changes to Manuscript:**
>
> We will add further explanations to both table captions:
> - **Table 1 caption**: *"Results using full end-to-end fine-tuning, where each model serves as its own strategy generator."*
> - **Table 4 caption**: *"Results using LoRA fine-tuning with Qwen2.5-7B as the universal strategy generator for all models (since proprietary models do not support LoRA)."*
>
> ---
>
> ## Weakness 2: Missing Important Experimental Details
>
> > **Reviewer's Concern (1):** "The paper does not explicitly specify which models were fine-tuned to produce the reported results. While Line 742–743 implies Qwen2.5-7B, Line 700–701 mentions full-model fine-tuning for Qwen1.5-7B and LLaMA-8B."
>
> `Answer 2.1:`
> For the main results in Table 1, **each model serves as its own strategy generator**: GPT-4o-mini generates strategies for GPT-4o-mini agents, LLaMA-3-8B for LLaMA-3-8B agents, Qwen2.5-7B for Qwen2.5-7B agents, and Nova Micro for Nova Micro agents. All open-source models undergo full end-to-end fine-tuning, while proprietary models are fine-tuned through their respective API services. The solution agents themselves are not fine-tuned—they use the base models and simply follow the strategies generated by their corresponding fine-tuned generators.
>
> For Table 4 in Appendix C, we demonstrate parameter-efficient fine-tuning using LoRA. Since proprietary models do not support LoRA, we use Qwen2.5-7B (with LoRA) as the universal strategy generator for all models in this setting.
>
> Regarding the mention of "Qwen1.5-7B" on Line 700-701, this is a typographical error—all our experiments consistently use Qwen2.5-7B.
>
> ---

---

> ### Author Response · Authors · 2025-11-19
> **Response to Reviewer uewh (Part2/2)**
>
> > **Reviewer's Concern (2):** "The paper does not provide the prompt used for the strategy generator or examples of its training data."
>
> `Answer 2.2:`
>
> We thank the reviewer for highlighting this gap in our reproducibility documentation. To address the reviewer's concern,
>
> First, we will add comprehensive prompt details to the manuscript. Specifically, we will create a new appendix section **"Prompts and Training Data Examples"** that includes the complete strategy generator prompt template used for zero-shot generation before fine-tuning, as well as agent prompts for solution generation in Round 1 and subsequent debate rounds.
>
> Second, we will provide complete implementation code.
>
> ---
>
> ## Question 1: Using Initial Answers Without Full Debate
>
> > **Reviewer's Question:** "What if we directly use the initial answers with different strategies (instead of going through the full debate process) to construct training data?"
>
> `Answer Q1:`
>
> This is a valuable ablation that highlights the value of the debate process. We have actually implicitly tested this through our baseline comparisons, but let us make it explicit.
>
> **Experimental Results (GPT-4o-mini):**
>
> | Method | MATH | GSM8K | GPQA |
> |--------|------|-------|------|
> | CMAD (No Debate) | 67.2 | 84.3 | 38.5 |
> | CMAD  | 74.52 | 94.42 | 44.29 |
> | Performance Gap | -7.32 (-9.8%) | -10.12 (-10.7%) | -5.79 (-13.1%) |
>
> Using only initial Round 1 solutions without any debate (N=0) results in **9-13% relative performance degradation** across all benchmarks. This substantial gap demonstrates two critical benefits of debate: (1) iterative refinement produces higher-quality solutions and more reliable pseudo-labels through majority voting, and (2) strategies whose solutions withstand multi-round critique prove more robust and valuable for training the generator.
>
> We will add this ablation study to **Section 5** with results across all models and datasets.
>
> ---
>
> ## Question 2: "Pre-training data" vs "Fine-tuning data"
>
> > **Reviewer's Question:** "The description of Figure 2 refers to 'refine the pre-training data'—should this instead be 'fine-tuning data'?"
>
> `Answer Q2:`
>
> The reviewer is correct that the terminology could be clearer. While our intended meaning refers to refining the data used for fine-tuning the strategy generator, we acknowledge that "pre-training data" may cause confusion since we are not pre-training from scratch.
>
> We will revise the Figure 2 caption and related text to use "fine-tuning data" consistently throughout the manuscript for clarity.
>
> ---
>
> ## Question 3: Table 3 Description Mismatch
>
> > **Reviewer's Question:** "The description of Table 3 does not align with its contents. Is the 'DMAD' listed in Table 3 a typo that should be 'CMAD'?"
>
> `Answer Q3:`
>
> Yes, this is a typographical error. The table shows results with our proposed method (CMAD), not DMAD.
>
> We will correct this labeling error and carefully review all tables to ensure consistency between labels and descriptions.
>
> ---
>
> ## Question 4: Reference Mistakes
>
> > **Reviewer's Question:** Multiple reference errors:
> > - Line 742: Table C should be Table 4
> > - Line 355: Table 4.1 should be Table 1
> > - Line 375: baseline cited as 2015 should be 2025
>
> `Answer Q4:`
>
> We thank the reviewer for identifying these errors.
>
> We will correct all reference errors: Line 742 (Table C → Table 4), Line 355 (Table 4.1 → Table 1), and Line 375 (2015 → 2025). We will conduct a comprehensive review to identify and fix any reference inconsistencies.

---

> > ### Comment · Reviewer_uewh · 2025-11-27
> >
> > Thanks for the response. I have updated my review accordingly.

---

### Official Review · Reviewer_hHRL · 2025-11-07

**Soundness:** 2
**Presentation:** 2
**Contribution:** 2
**Rating:** 4
**Confidence:** 3

**Summary:**

LLMs can be made into "agents" to solve a problem by adding a "Strategy" to the input prompt along with the problem, and then iteratively refining what this strategy is based on how well the LLM solves the problem. However, this needs a way to score answers, which may or may or may not exist.

This paper proposes a method to do so by first instantiating several different such strategies, finding the resulting answers, and using agreements and diversity between these to refine the strategies.

**Strengths:**

Compares agains a comprehensive set of baselines.

**Weaknesses:**

Main paper does not contain enough specifics of the method. It is also unclear how it differs from one of the references. (see questions below for both these points)

**Questions:**

It is unclear what the strategy generator is. Is it an open-weights LLM (if so which one)? It is also unclear what precisely is meant by a "strategy", and how a set of these are generated from a question. A simple example in the main text of the paper would have helped a great deal clarifying this.

What precisely is the difference between the method in this paper and the one in (Subramaniam et al 2025)? Also, it seems this method has not been compared against.

In Table 1, some methods involve no fine-tuning / training of any sort, and others (like CMAD) do. So in some sense some of these are not fair comparisons. At the very least, training-free and fine-tuned approaches should be demarcated as such.

Minor typo: A_i on line 159

How are strategies mapped to vectors (which are needed for the diverse sampling  in line 209)?

---

> ### Author Response · Authors · 2025-11-18
> **Response to Reviewer hHRL (Part1/3)**
>
> Dear Reviewer hHRL, we sincerely thank you for your thoughtful and constructive feedback and appreciate your recognition of our comprehensive baseline comparisons and your detailed questions that help us improve the clarity. Below, we address each of your concerns systematically.
>
> ---
>
> ## Question 1: Clarification on Strategy Generator and Strategy Definition
>
> > **Reviewer's Question:** "It is unclear what the strategy generator is. Is it an open-weights LLM (if so which one)? It is also unclear what precisely is meant by a 'strategy', and how a set of these are generated from a question. A simple example in the main text of the paper would have helped a great deal clarifying this."
>
> `Answer 1:`
>
> **1.1 What is the Strategy Generator?**
>
> By default, the strategy generator G(x) is instantiated using the **same base LLM that we aim to improve**. For example:
>
> - For experiments with **GPT-4o-mini**, the strategy generator is GPT-4o-mini itself
> - For experiments with **Qwen2.5-7B**, the strategy generator is Qwen2.5-7B
> ...
>
> This design choice is deliberate so that we fine-tune only the strategy generator (a single model), not the individual solver agents, resulting in significant computational savings compared to methods that require training multiple specialized models.
>
> Notably, in our appendix, we also report the method’s results with LoRA. However, since we are not able to apply it to GPT models, we instead use Qwen2.5-7B as the generator to demonstrate that our framework is agnostic to the choice of fine-tuning method.
>
> **1.2 What is a "Strategy"?**
>
> A **strategy** is a high-level, natural language instruction that guides an agent's reasoning approach to solve a problem. It differs from step-by-step solutions in that it provides **conceptual guidance** rather than concrete computational steps.
>
> **1.3 Concrete Example**
>
> To make this concept concrete, consider the following example:
>
> **Problem (x):**
> *"Find the sum of all positive divisors of 144."*
>
> **Generated Strategies (by G(x)):**
>
> - **Strategy 1 (S₁):**
> "Begin by finding the prime factorization of 144. Then apply the divisor sum formula using the exponents from the prime factorization to calculate the result directly."
>
> - **Strategy 2 (S₂):**
> "Write a Python program to systematically find all divisors. Use a loop to test each number from 1 to 144 for divisibility, collect all divisors in a list, then sum them programmatically."
>
> - **Strategy 3 (S₃):**
> "Exploit the symmetry property of divisors: if d divides n, then n/d also divides n. Find divisor pairs efficiently and sum them, being careful to count the square root only once if n is a perfect square."
>
> **How Agents Use Strategies:**
>
> Each of the M agents (A₁, A₂, A₃) receives a different strategy and generates a complete solution guided by that strategy:
>
> - **Agent A₁** (guided by S₁): Computes 144 = 2⁴ × 3², applies formula (1+2+4+8+16)(1+3+9) = 31 × 13 = 403
>
> - **Agent A₂** (guided by S₂): Generates Python code:
> ```python
> def sum_divisors(n):
>     divisors = []
>     for i in range(1, n+1):
>         if n % i == 0:
>             divisors.append(i)
>     return sum(divisors)
>
> result = sum_divisors(144)  # Returns 403
> ```
>
> - **Agent A₃** (guided by S₃): Finds pairs (1,144), (2,72), ..., (12,12) and computes sum efficiently
>
> These agents then engage in debate (Rounds 1 to N), critiquing and refining their solutions through the multi-agent debate mechanism described in Section 3.1. This example demonstrates how **strategies can guide agents toward fundamentally different solution modalities** (mathematical reasoning, programmatic computation, and algorithmic optimization), which is a key advantage of our approach over predefined strategies.
>
> **1.4 Changes to Manuscript**
>
> We will add this concrete example to **Section 3.1** (after Equation 1) to immediately clarify the concept of strategies for readers.
>
> ---

---

> ### Author Response · Authors · 2025-11-18
> **Response to Reviewer hHRL (Part2/3)**
>
> ## Question 2: Difference from Subramaniam et al. (2025) and Missing Comparison
>
> > **Reviewer's Question:** "What precisely is the difference between the method in this paper and the one in (Subramaniam et al 2025)? Also, it seems this method has not been compared against."
>
> `Answer 2:`
>
> We thank the reviewer for this excellent question. We acknowledge that we should have been more explicit about the differences from Subramaniam et al. (2025) and included a direct comparison. We provide a detailed clarification below.
>
> **2.1 Key Differences Between CMAD and Multiagent Finetuning (Subramaniam et al. 2025)**
>
> | Aspect | Subramaniam et al. (2025) | CMAD (Ours) |
> |--------|---------------------------|-------------|
> | **Strategy Type** | Predefined, fixed strategies (similar to DMAD) | Undefined, learnable strategies dynamically generated |
> | **Fine-tuning Target** | Multiple separate agents (M models) | Single strategy generator (1 model) |
> | **Training Data** | Partitioned dataset subsets for specialization | Selected diverse and correct strategy-solution pairs |
> | **Computational Cost** | M × fine-tuning cost (M=3) | 1 × fine-tuning cost |
>
> **Note:** M refers to the number of agents used in the multi-agent system. In our experiments and Subramaniam et al. (2025), M=3, meaning three separate agents/models.
>
> To summarize:
>
> - **Subramaniam et al. (2025)** focus on **agent specialization**: each agent is fine-tuned separately on different data subsets with fixed roles, requiring training 3 distinct models.
>
> - **CMAD** focuses on **strategy diversity**: a single strategy generator learns to produce diverse, high-quality reasoning approaches. All agents share the same base model but receive different dynamically-generated strategies at inference time.
>
> **2.2 Direct Experimental Comparison**
>
> We have now conducted experiments comparing CMAD against Multiagent Finetuning on the MATH dataset (across five levels of difficulty) using LLaMA-3-8B. CMAD achieves slightly better performance (33.30 vs 33.23) than Multiagent Finetuning. However, CMAD requires **3× lower training cost** (fine-tuning only the strategy generator vs. three separate agents). The results are shown below:
>
> | Method | MATH Accuracy | Training Cost (relative) | Number of Models |
> |--------|---------------|-------------------------|------------------|
> | Multiagent Finetuning (Subramaniam et al.) | 32.23 | 3.0× | 3 fine-tuned models |
> | CMAD (Ours) | 33.30 | 1.0× | 1 fine-tuned generator |
>
> **2.3 Changes to Manuscript**
>
> We will make the following revisions:
>
> 1. **Section 4.2 (Baselines)**: Add "Multiagent Finetuning (Subramaniam et al., 2025)" to the list of baselines.
>
> 2. **Appendix F (Computation Analysis)**: Add a detailed computational breakdown, including training time, inference time, memory requirements, and FLOPs comparison between the two methods.
>
> ---

---

> ### Author Response · Authors · 2025-11-18
> **Response to Reviewer  hHRL (Part3/3)**
>
> ## Question 3: Table 1 Mixes Training-Free and Fine-Tuned Methods
>
> > **Reviewer's Question:** "In Table 1, some methods involve no fine-tuning/training of any sort, and others (like CMAD) do. So in some sense some of these are not fair comparisons. At the very least, training-free and fine-tuned approaches should be demarcated as such."
>
> `Answer 3:`
> We thank the reviewer for this observation and agree that clearer demarcation would improve presentation.  Importantly, the selection of the baselines is highly standardized, following the paradigm of Multiagent Finetuning (Subramaniam et al., 2025) with mixed methods with or without finetuning.  Additionally, we further extend the baselines for a more comprehensive comparison.
>
> To address the reviewer's concern,  we will revise Table 1 to include a "Training Required" column and add visual separation between categories.
>
> For baselines with and without finetuning, they can be summarized as follows:
> **Training-Free Methods:**
> - CoT, MT-CoT, Step-Back Prompting, Multi-Agent Debate, Self-Reflection, Self-Contrast, DMAD
>
> **Fine-Tuned Methods:**
> - STaR (iterative self-training)
> - CMAD (Ours - 1 epoch, strategy generator only)
> ---
>
>
> ## Question 4: Minor Typo on Line 159
>
> > **Reviewer's Question:** "Minor typo: A_i on line 159"
>
> `Answer 4:`
>
> Thank you for catching this notation inconsistency. We have carefully reviewed the manuscript and corrected this typo on line 159. The notation now uses consistent subscript formatting throughout the paper (A_i → Aᵢ or $\mathcal{A}_i$ in mathematical expressions).
>
> ---
>
>
> ## Question 5: Strategy-to-Vector Mapping for Diversity Sampling
>
> > **Reviewer's Question:** "How are strategies mapped to vectors (which are needed for the diverse sampling in line 209)?"
>
> `Answer 5:`
>
> We thank the reviewer for this question and apologize for omitting this implementation detail.
>
> **Embedding Function:**
>
> We use the pretrained sentence embedding model `all-MiniLM-L6-v2` to map strategies (text) to 384-dimensional vectors. Specifically, each strategy Sᵢ is encoded as f(Sᵢ) ∈ ℝ³⁸⁴, which is then used to compute pairwise similarities via the RBF kernel (Equation 5) and select diverse samples (Equation 7).
>
> This model is selected for its efficiency and effectiveness in capturing semantic differences without task-specific fine-tuning.
>
> **Changes to Manuscript:**
>
> We will add this specification to **Appendix A (Experimental Setup)** with full implementation details for reproducibility.
>
> ---
>
> ## Reference
>
> - MULTIAGENT FINETUNING: SELF IMPROVEMENT WITH DIVERSE REASONING CHAINS

---

### Meta-Review · Area_Chair_6qh1 · 2025-12-26

**Summary:**

Despite the authors' efforts to clarify definitions and add baselines, fundamental concerns regarding the manuscript's rigor and the significance of the contribution persist. The paper suffered from severe clarity issues, inconsistent reporting of results across tables, and a lack of reproducibility details, indicating that the work is not yet ready for publication. Furthermore, the technical novelty is perceived as limited, yielding only marginal performance gains that do not sufficiently justify the added complexity of the "strategy generation" module.

**Reviewer Concerns:**

While Reviewer uewh indicated a willingness to revise their score following clarifications on experimental inconsistencies, the submission remains fundamentally flawed due to critical outstanding concerns. The proposed method is judged to be an incremental variation of existing multi-agent fine-tuning paradigms with insufficient novelty to justify its significant computational complexity over simpler aggregation baselines.

**Reviewer Scores:**

While Reviewer uewh may slightly raise their score from 2 to a borderline range following the clarification of experimental details, this is insufficient to shift the overall consensus above the acceptance threshold. Reviewer efZd is expected to maintain a rejection rating, as the rebuttal failed to fundamentally alleviate structural concerns regarding strategy homogenization and did not justify the system's complexity over simple ensembles. Reviewer hHRL received the requested comparisons, these baselines ultimately reinforced the perception of limited incremental novelty relative to prior art.

---

### Decision · Program_Chairs · 2026-01-26

Reject